# The Impact of Nasal Patency on Vocal Fold Nodule Formation in Children

**DOI:** 10.3390/jcm14134743

**Published:** 2025-07-04

**Authors:** Aleksander Zwierz, Krzysztof Domagalski, Krystyna Masna, Paweł Burduk

**Affiliations:** 1Department of Otolaryngology, Phoniatrics and Audiology, Faculty of Medicine, Ludwik Rydygier Collegium Medicum, Nicolaus Copernicus University, 75 Ujejskiego Street, 85-168 Bydgoszcz, Poland; krymasna@gmail.com (K.M.); pburduk@wp.pl (P.B.); 2Department of Immunology, Faculty of Biology and Veterinary Science, Nicolaus Copernicus University, 1 Lwowska Street, 87-100 Torun, Poland; krydom@umk.pl

**Keywords:** dysphonia, vocal fold nodules, children, adenoid, adenoid hypertrophy, turbinate hypertrophy

## Abstract

**Objectives**: This study aimed to endoscopically assess nasal patency in terms of adenoid obstruction and its mucous coverage, as well as nasal obstruction caused by the inferior nasal turbinate in children with vocal fold nodules. **Methods**: A retrospective study was conducted involving 54 children admitted to an ENT clinic due to hoarseness caused by vocal fold nodules from 2022 to 2024. The study analyzed medical history, the results of performed flexible nasofiberoscopy and tympanometry. **Results**: Children with vocal fold nodules snored and slept with open mouths less frequently than the control group of other patients admitted to the ENT outpatient clinic without voice disorders (*p* = 0.003 and 0.004, respectively). Pathological mucous coverage of the adenoid was observed more often (*p* = 0.02). The mean adenoid size in the A/C ratio was 52.1% compared to 63.4% in the control group (*p* = 0.01). **Conclusions**: Children with vocal fold nodules typically have smaller adenoids, fewer incidents of snoring and open-mouth breathing, but more frequent pathological nasal mucus. It was not possible to prove that the incorrect breathing path through the mouth, causing reduced humidity of the inhaled air, affects the formation of vocal fold nodules.

## 1. Introduction

Vocal nodules may be present in 35% to 78% of children suffering from voice disorders [1]. The genesis of vocal nodules, which is a common cause of hoarseness and dysphonia in children, has been examined from sociological, psychological, and pathophysiological perspectives. Numerous studies confirm that their formation is favored by excessive vocal activity, especially in noisy environments, during sports activities, in large families, or among children with younger siblings. Organic causes include reflux, allergy, and dietary factors such as the consumption of carbonated and junk food [2,3]. Many clinicians suggest that breathing dry, dehumidified air through the mouth, often caused by impaired nasal patency, can also contribute. These voice disorders typically begin in preschool children, when nasal patency can be affected by nasal inferior turbinate hypertrophy (ITH) and adenoid hypertrophy (AH). The aim of this study was to endoscopically assess nasal patency concerning adenoid obstruction and mucous coverage, as well as nasal obstruction by the inferior nasal turbinate in children with vocal fold nodules.

## 2. Materials and Methods

### 2.1. Study Group

A retrospective study involved 61 children admitted to the ENT clinic from 2022 to 2024 for hoarseness persisting for more than three months, during which flexible nasofiberoscopy and laryngeal evaluation were performed and pathological conditions on vocal folds were stated. Two children were excluded due to polyps, and one child had a cyst on the vocal fold. Thus, 58 children with vocal fold nodules were included in the initial analysis. One child with a history of cleft palate surgical treatment and three children who had undergone adenoidectomy were ultimately excluded, leaving 54 children (15 girls and 39 boys, aged 3 to 9 years) in the study. The control group comprised 66 age- and sex-matched children who visited the ENT outpatient clinic for other reasons (hypoacusis, adenoid hypertrophy or middle ear effusion suspicions, rhinitis, or recurrent upper respiratory tract infections) without any voice disorders. To account for the potential influence of seasonal variations on nasal mucus and dysphonia, the control group was selected to match the group with vocal fold nodules in terms of study season [4,5,6]. To simplify the comparison of our study to others performed in different regions and climates, we adopted two main thermal seasons—winter and summer—that were considered to be the cutoff point for a mean temperature of 10 °C.

### 2.2. Exclusion Criteria

Children with craniofacial malformations, genetic disorders (e.g., Down syndrome); nasal septal deviation; nasal polyps; active upper respiratory infections; previous nasal, cleft palate, or adenoid surgery; or other causes of hoarseness were excluded.

### 2.3. Diagnostics

A comprehensive medical history was collected for each child, including symptoms reported by parents such as hoarseness, snoring, sleeping with the mouth open, allergies, and asthma.

All children underwent flexible nasopharyngeal endoscopy, assessing the inferior nasal turbinate and measuring the free space for breathing in the common nasal meatus at the level of the inferior nasal turbinate. In the performed endoscopic procedure, both nasal cavities were initially assessed by inserting the endoscope into the nasal vestibule. However, full nasopharyngoscopy was performed on the more patent side to minimize pain or discomfort for the patient. These examinations were conducted by a pediatric otorhinolaryngologist (AZ) using the Karl Storz (Tuttlingen, Baden Württemberg, Gremany) Tele Pack endoscopic system, equipped with a flexible nasopharyngoscopy tool (2.8 mm outer diameter, 300 mm length). Due to the nasal cycle, which involves alternating temporary swelling of the mucous membrane in one of the nasal cavities, the patency of the less-obstructed nasal cavity was assessed. Recognizing the absence of a standardized classification for assessing inferior nasal turbinate hypertrophy in children, we divided the children into two groups based on the ratio of space occupied by the inferior nasal turbinate to the total airway space—up to 50% and 50% or more [7,8]. The rhinoscopy assessment was deemed more objective, with correlations between the clinical scores of nasal obstruction and Peak Nasal Inspiratory Flow (PNIF) being demonstrated by various authors [9].

Additionally, the adenoid size and its mucus coverage were assessed. Previous studies have indicated a clinically significant cut-off value for adenoid size measured using the adenoid/choana (A/C) ratio, with a threshold of 75% [10]. Patients were thus analyzed based on whether the adenoid constituted less than 75% or more than 75% of the nasopharyngeal surface area. The degree of adenoid mucus coverage visible in endoscopic examination was also assessed, as the presence and constitution of mucus can affect nasal patency [11]. The Mucus of Adenoid Scale by Nasopharyngoscopy Assessment (MASNA) was used to describe the amount of mucus covering the adenoid on a 4-point scale, as follows: 0 (no mucus), 1 (residue of clear watery mucus), 2 (some dense mucus), and 3 (copious thick dense mucus) [5].

For children with hoarseness persisting for more than three months, flexible laryngeal endoscopy was performed through the nose to diagnose laryngeal conditions. The postnasal discharge observed on the posterior pharyngeal wall, features of laryngopharyngeal reflux (LPR), and the presence of pathological changes on the vocal folds were evaluated. Vocal nodules were defined as bilateral sessile lesions, clear and symmetric, located on the free border of the vocal folds at the junction between the anterior third and the middle third of the glottic phonation area. Similarly to Akif Kiliç we distinguished three types of vocal fold nodules, as follows: type 1—minimal lesion; type 2—immature nodules; and type 3—mature nodules. Localized edema or irregularity at the junction of the anterior- and middle-third of the vocal folds was accepted as a minimal lesion; hyperemic, edematous, fusiform lesions were classified as being immature; and fibrotic, whitish lesions were classified as mature nodules [1].

Tympanometry was conducted using the GSI (Eden Prairie, MN, USA) 39 AutoTymp TM from Grason-Stadler, with the results being classified according to the Liden and Jerger classification system for tympanograms. Each patient’s right and left ear tympanograms were recorded; however, for simplicity, the results were divided into three groups based on the worst tympanogram in either ear, as follows: type B (considered the worst), type C (worse), and type A (indicative of normal function).

### 2.4. Statistical Analysis

For descriptive statistics, quantitative variables were presented as means ± standard deviation (SD), while categorical variables were summarized using frequency counts and percentages. Statistical significance was determined using the Chi-square method or Fisher’s exact test for categorical variables, while Student’s *t*-test or one-way ANOVA was used for quantitative variables to assess differences between study groups.

Variables significantly related to vocal fold nodules in the univariate analysis were included in the logistic regression analysis to identify independent prognostic factors useful in assessing vocal fold nodules in ENT patients. Odds ratios (ORs) and 95% confidence intervals (95% CIs) were also calculated for considered clinical variables in regression models. For all these tests, two-tailed *p*-values were used, and differences at the level of *p* < 0.05 were considered significant. All statistical analyses were performed with SPSS (Statistical Package for the Social Sciences, version 28, Armonk, NY, USA) software.

### 2.5. Ethics

Ethical approval for this study was obtained from the ethics committee of Nicolaus Copernicus University (KB 141/2022).

## 3. Results

The mean age of the children with vocal fold nodules without voice disorders was 5.2 years, compared to 5.1 years in the control group. Both groups had a similar distribution across gender and the period of endoscopic examination in the thermal seasons. However, flexible endoscopy revealed less children in the vocal fold nodule group with significant nasopharyngeal obstruction by the adenoid (A/C ratio ≥ 75%). In relation to this, this group differed significantly from the control group (*p* = 0.01). Moreover, the mean adenoid size in the A/C ratio in the group with vocal fold nodules was 52.1%, but was 63.4% in the control group. That was a statistically significant difference (*p* = 0.003). The analyzed group of patients with vocal nodules statistically significantly differed from the control group both in terms of symptoms of snoring reported by parents and sleeping with the mouth open (*p* = 0.003 and *p* = 0.004, respectively). The children in the control group snored and slept with their mouths open more often. A larger number of patients with vocal nodules exhibited degrees of nasal mucous coverage (on the MASNA scale), indicating the presence of pathological secretion, which differed significantly from the control group (*p* = 0.02).

In the studied group of patients with vocal nodules, endoscopic examination revealed features of laryngopharyngeal reflux in almost 74.1%, and postnasal discharge in 94.4%. In total, 11% of these children were treated for asthma, 31.5% underwent allergy tests, and almost 65% of these tests were positive; however, 68.5% of the children were not tested for allergies at the time of vocal nodule diagnosis. Similarly, in the control group, 62.1% of the children were not subjected to allergy tests. When analyzing nasal obstruction (≥50% total nasal airway space occupied by the inferior turbinate) and tympanometry results, there was no difference between the groups (Table 1).

Regression analysis for factors such as an adenoid size of less than 75% A/C ratio, the presence of any pathological nasal secretion (1 to 3 degrees on the MASNA scale), and the absence of snoring showed that these factors increase the risk of vocal nodules in ENT patients (*p* = 0.031, 0.004, and 0.027, respectively). The risk of vocal nodules increases 3-fold with an adenoid hypertrophy of less than 75% A/C ratio, 4-fold with the presence of pathological nasal discharge (MASNA 1 to 3 degrees), and almost 3fold in the absence of snoring, when compared to the studied control group (Table 2).

## 4. Discussion

Our study did not show that nasal blockage by the adenoid, which correlates to significant adenoid hypertrophy (A/C ratio ≥ 75%), promotes vocal nodule formation. This adenoid size, exceeding 75% in A/C ratio, was found to significantly affect the deterioration of nasal patency and the resolution of related ailments in children [10]. In the group of children with vocal fold nodules, there was a prevalence of children with relatively small adenoids that were not eligible for surgical treatment. This is corroborated by the medical history provided by the parents of these children, who reported fewer symptoms of snoring and open-mouth sleeping compared to the control group. This result is puzzling to us, because this finding suggests that smaller adenoids may promote the formation of nodules. This phenomenon may be related to the often-overlooked role of the adenoid in producing supraglottal pressure in children. Aronsson et al. observed increased subglottal pressure in patients with vocal fold nodules compared to healthy speakers to achieve the same voice quality [12]. The adenoid, in conjunction with the soft palate, helps separate the nasal and oral cavities, ensuring proper airflow through the nose and mouth, which is crucial for acceptable speech [13]. However, Maryn et al. also noted that adenoid hypertrophy (AH) could cause hyponasality or even denasality [14]. The term veloadenoidal closure, increasingly used by many authors for preschool children, underscores the significance of adenoid size in the phonation process. A small adenoid may reduce veloadenoidal closure, forcing an increase in subglottic pressure [15]. This could be analogous to the increased pressure and force stress seen in regions with higher incidences of laryngeal cancer, which is linked to the penetration of inhaled toxic agents due to increased pressure and reduced air velocity [16]. Therefore, the optimal adenoid size might be crucial for phonation in children, warranting further research to determine the appropriate size for this process. In this context, it is also puzzling that in the entire study group of children with vocal nodules, three children’s nodules were stated later after the earlier performed adenoidectomy, and one of the children underwent soft palate repair surgery. Certainly, this group is too small to draw any further conclusions.

The study demonstrated a significant impact of nasal secretions and postnasal discharge on the formation of vocal nodules. Turley et al. confirmed that patients with rhinitis, both allergic and non-allergic, had a higher prevalence of dysphonia. Patients with dysphonia in the described study were more likely to report postnasal discharge, dry throat, or globus sensation [3]. Allergies can exacerbate postnasal discharge, thereby contributing to the formation of vocal nodules or dysphonia [17,18,19]. The flowing dense secretion covering the vocal folds can cause excessive strain on them and increase coughing. Such actions have a negative effect on the vocal folds. De Bodt et al. identified allergic dysphonic girls as being the highest-risk group for persistent adolescent vocal fold nodules [20]. Randhaw et al. suggested that the role of allergies in the development of dysphonia might be more significant than that of laryngopharyngeal reflux (LPR) [21].

Several studies have shown a relationship between allergies and hypertrophy of the nasal turbinate [22,23,24]. Hamizan et al. indicated that the swelling of the inferior turbinate head might predict allergies, although no relationship was found between inferior turbinate tail hypertrophy and allergies [22]. Karabutut et al. demonstrated that both the hypertrophy and color of the inferior turbinate could indicate a higher likelihood of allergies [23]. These studies were conducted on adults, while allergy tests in preschool children are less common. Ciprandi’s work is among the few studies that have shown a relationship between inferior turbinate hypertrophy (ITH) and allergies in children [24]. Unfortunately, in our study, we were unable to demonstrate a relationship between the size of the nasal turbinate and the occurrence of vocal nodules. Perhaps this is related to the incorrectly adopted scale and cut-off point at the level of 50% of nasal obstruction by the inferior nasal turbinate for the analyzed groups, but there are still no studies and classifications in this area clearly indicating what level of nasal obstruction by the inferior nasal turbinate has a significant impact on its patency in children [7]. This is particularly difficult in the case of preschool and early school children, because at this age, the patency of the nose depends on both the inferior nasal turbinates’ hypertrophy and the adenoid size and its mucous coverage.

In our study, most children were not tested for allergies (68.5% and 62.1%, respectively), but the presence of pathological nasal secretion, which was significantly more common in children with vocal nodules, might indirectly indicate allergies. Taking into account the children examined for allergy in the analyzed group and the control group, it was significantly more common in the group of children with vocal nodules (65% vs. 48%).

## 5. Conclusions

Children with vocal fold nodules typically have smaller adenoids compared to other ENT patients, but they report more frequent pathological nasal mucus. Snoring and open-mouth breathing is not related to vocal cord nodules. An adenoid size of less than 75% A/C ratio, the presence of pathological nasal mucus, and the absence of snoring are factors that increase the risk of vocal nodules in ENT patients. The risk of vocal fold nodules increases three times with an adenoid hypertrophy of less than 75% A/C ratio, as well as increasing four times with the presence of pathological nasal discharge (MASNA 1 to 3 degrees). In the course of the conducted research, it was not possible to prove that the incorrect breathing path through the mouth, causing reduced humidity of the inhaled air, affects the formation of vocal fold nodules. Further research on the relationship between anatomical conditions and pathophysiological processes in vocal nodules formation is warranted.

## Figures and Tables

**Table 1 jcm-14-04743-t001:** Characteristics of the study groups.

Characteristic		Children with Vocal Fold Nodules (*n* = 54)	Control Group (Without Vocal Fold Nodules) (*n* = 66)	*p*-Value
Gender	female	15 (27.8%)	16 (24.2%)	0.660
	male	39 (72.2%)	50 (75.8%)	
Age (years)	mean ± SD	5.2 ± 1.6	5.1 ± 1.3	0.685
Thermal season	summer	24 (44.4%)	23 (34.8%)	0.348
	winter	30 (55.6%)	43 (65.2%)	
Adenoid size (A/C ratio)	mean ± SD	52.1 ± 20.7	63.4 ± 20.3	**0.003**
	<75%	45 (83.3%)	41 (62.1%)	**0.010**
	≥75%	9 (16.7%)	25 (37.9%)	
Adenoid mucus coverage (MASNA scale)	0	8 (14.8%)	22 (33.3%)	**<0.001**
	1	14 (25.9%)	18 (27.3%)	
	2	29 (53.7%)	13 (19.7%)	
	3	3 (5.6%)	13 (19.7%)	
Adenoid mucus coverage (MASNA scale) (normal vs. pathological)	0	8 (14.8%)	22 (33.3%)	**0.020**
	1–3	46 (85.2%)	44 (66.7%)	
Part of the total nasal airway space occupied by inferior turbinate	mean ± SD	56.4 ± 12.6	55.8 ± 12.4	0.809
	<50%	12 (22.2%)	20 (30.3%)	0.319
	≥50%	42 (77.8%)	46 (69.7%)	
Type of tympanogram	AA	38 (70.4%)	40 (60.6%)	0.486
	AB/BA	3 (5.6%)	3 (4.5%)	
	AC/CA	5 (9.3%)	5 (7.6%)	
	BB	3 (5.6%)	10 (15.2%)	
	BC/CB	0 (0.0%)	2 (3.0%)	
	CC	5 (9.3%)	6 (9.1%)	
Tympanogram (worst)	A	38 (70.4%)	40 (60.6%)	0.249
	C	10 (18.5%)	11 (16.7%)	
	B	6 (11.1%)	15 (22.7%)	
Tympanogram (normal vs. pathological)	A	38 (70.4%)	40 (60.6%)	0.265
	non A	16 (29.6%)	26 (39.4%)	
Snoring	yes	8 (14.8%)	24 (36.4%)	**0.003**
	periodic	13 (24.1%)	21 (31.8%)	
	no	33 (61.1%)	21 (31.8%)	
Open-mouth sleeping	yes	14 (25.9%)	37 (56.1%)	**0.004**
	periodic	21 (38.9%)	17 (25.8%)	
	no	19 (35.2%)	12 (18.2%)	
Allergy	yes	11 (20.4%)	12 (18.2%)	0.44
	no	6 (11.1%)	13 (19.7%)	
	not tested	37 (68.5%)	41 (62.1%)	
Types of vocal folds nodules	minimal lesion	31 (57.4%)	N/A	-
	immature nodules	19 (35.2%)	N/A	
	mature nodules	4 (7.4%)	N/A	
Postnasal discharge	yes	51 (94.4%)	N/A	-
	no	3 (5.6%)	N/A	
Laryngopharyngeal reflux (LPR)	yes	40 (74.1%)	N/A	-
	no	14 (25.9%)	N/A	
Asthma	yes	6 (11.1%)	N/A	-
	no	48 (88.9%)	N/A	

A/C ratio: adenoid-to-choana ratio, N/A: not avaliable.

**Table 2 jcm-14-04743-t002:** Logistic regression analysis for the prediction of vocal fold nodules in children aged 3–9 years.

Characteristic	*p*-Value	OR	95% CI
Adenoid size (A/C ratio), ≥75%	**0.031**	**0.35**	**0.14–0.91**
Adenoid mucus coverage (MASNA scale), 1–3 (pathological)	**0.004**	**4.30**	**1.58–11.71**
Snoring, yes	**0.027**	**0.37**	**0.15–0.89**

## Data Availability

Additional data supporting the reported results may be available upon reasonable request.

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
