# Peer review of "The Impact of Nasal Patency on Vocal Fold Nodule Formation in Children"

_jcm, 2025, doi:10.3390/jcm14134743_

Round 1
Reviewer 1 Report
Comments and Suggestions for Authors A well-designed study to show whether nasal breathing and the changes that occur in the airway affect the appearance of vocal fold nodules. The mechanism of the formation of nodules on the vocal cords in children is motor hyperactivity, which includes increased activity of the vocal muscles due to increased volume, increased volume of speech, poor voice placement. Changes in the vocal cords occur due to mechanical trauma. These changes occur in the middle of the vocal cords, and not at the junction of the front and middle thirds, because the vocal tract in children is small, so the entire vocal cord is membranous, and the punctum is the maximum vibration in the middle. In the beginning, hoarseness occurs due to mechanical trauma, and later the appearance of growths and inability to fold the vocal cords leads to hoarseness. There is no sentence„The mean age of the children with vocal fold nodules without voice disorders was 5.2 years compared to 5.1 years in the control group.„
Minor corrections are necessary. Why is there more secretion on the adenoid in children with nodules on the vocal cords? That is a question that needs to be considered and tried to prove. Is there insufficient air flow through the nose with this hyperactivity because they use the oral route of air flow more? There is a connection to be seen. Greater airflow through the nose leads to more evaporation of secretions and drying of the mucous membrane.
Author Response
Der Reviewer,
Thank you very much for the huge amount of work put into evaluating my work and trying to make it better. Below I will try to give a precise answer to the questions posed:
Minor corrections are necessary. Why is there more secretion on the adenoid in children with nodules on the vocal cords? That is a question that needs to be considered and tried to prove.
Thank You for this question.This issue is taken up by some researchers and clinicians. The thick mucus from the adenoid, post nasal drip flowing down the posterior pharyngeal wall, may settle on the vocal folds and burden them and cause additional vocal effort. Allergy rhinitis may be important in this regard, but more than 60% of children in the study have not been tested for it. Therefore, some clinicians recommend the inclusion of antihistamine drugs in young patients treated for vocal nodules.
In addition, we would like to point out that in our previous studies we showed a correlation between adenoid mucus and the thermal season. To exclude this relationship, the control group was selected so that it did not differ statistically significantly from the study group in terms of the season.
Is there insufficient air flow through the nose with this hyperactivity because they us
e the oral route of air flow more? There is a connection to be seen. Greater airflow through the nose leads to more evaporation of secretions and drying of the mucous membrane.
Yes, this is a very accurate observation and a relationship worth analyzing, but in the group of children with vocal nodules, in whom there was a smaller adenoid and probably more often a nasal inspiratory flow, a greater degree of mucus of on adenoid was observed.
Thank You
Reviewer 2 Report
Comments and Suggestions for Authors
This is a well-written manuscript that studies an interesting idea: whether nasal obstruction with mouth breathing has an impact on vocal cord nodules in children.
- Among many factors that can contribute to the vocal cord nodules in children the excessive voice activity probably is the most important. It is not easy in a retrospective study where, for one group of children, there are probably no records regarding voice abuse to have a weighted sample.
- In line 75, page 2/6, it is mentioned that the patency of the less obstructed nasal cavity was assessed. Did all children have bilateral endoscopy?
- If the children have bilateral endoscopy why do authors include only the most obstructed nasal cavity and not a total measurement of both the nasal cavities?
- Line 132, page 3/9: I suppose the authors mean….. “That was a statistically significant difference (p = 0.003)”.
- If the secretions on the adenoids in the group of children with nodules, where in the context of allergy or GER, the secretions probably do not constitute an independent factor for the vocal nodules.
- Since the majority of the children in the study group have allergies, it is essential to have information regarding chronic cough and the use of steroid inhalers, as these are also parameters implicated in the formation of vocal cord nodules.
Even if the authors provide an explanation implicating subglottic pressure in the discussion section, it is not rational to relate nodules to the absence of snoring. Snoring is a pathological sign. It would be more rational if the conclusion was that snoring is not related to vocal cord nodules, which would be interesting.
- To study the hypothesis that “a small adenoid may reduce veloadenoidal closure, forcing an increase in subglottic pressure,” the authors should better compare two random groups of children with adenoid hypertrophy and without any vocal cord lesions. In this study, where one group of children already has nodules, the results are biased.
- In line 180, page 5/9, what is the optimal adenoid size? Would it be normal for a child of about 5 years old to have no adenoids at all?
Author Response
Der Reviewer,
Thank you very much for the huge amount of work put into evaluating my work and trying to make it better. Below I will try to give a precise answer to the questions posed
- In line 75, page 2/6, it is mentioned that the patency of the less obstructed nasal cavity was assessed. Did all children have bilateral endoscopy?
Yes, In the performed endoscopic procedure in children, both nasal cavities were initially assessed by inserting the endoscope into the nasal vestibule. However, full nasopharyngoscopy was performed on the more patent side to minimize pain or discomfort for the patient. We ‘ve added it in the text.
- If the children have bilateral endoscopy why do authors include only the most obstructed nasal cavity and not a total measurement of both the nasal cavities?
As mentioned in the text, it is very difficult to assess the size of the inferior nasal turbinates in children in order to assess their effect on nasal patency. There are no appropriate classifications that would be useful in this age group. Initially, we used the classification proposed by Camacho (Camacho M, Zaghi S, Certal V, Abdullatif J, Means C, Acevedo J, Liu S, Brietzke SE, Kushida CA, Capasso R. Inferior turbinate classification system, grades 1 to 4: development and validation study. Laryngoscope. 2015 Feb; 125(2):296-302. doi: 10.1002/lary.24923. Epub 2014 Sep 12 25215619) however, we also did not obtain statistically significant differences in both groups, Analyzing the ROC curve, it was not possible to determine the cut-off point and the most significant results were obtained in the area of Camacho 50% and this value was adopted in the study.
In the paper, the patency of the nasal cavity was analyzed, after which the smaller inferior nasal turbinate was found, because this side shows the maximum nasal patency, and in our opinion, additional description of the less permeable side would make it difficult to assess the already difficult one. We have written it in the line 69 that we analyzed less obstructed side” Due to the nasal cycle, which involves alternating temporary swelling of the mucous membrane in one of the nasal cavities, the patency of the less obstructed nasal cavity was assessed”.
- Line 132, page 3/9: I suppose the authors mean….. “That was a statistically significant difference (p = 0.003)”.
Thank You we ‘ve corrected it in the text
- If the secretions on the adenoids in the group of children with nodules, where in the context of allergy or GER, the secretions probably do not constitute an independent factor for the vocal nodules.
This issue is taken up by some researchers and clinicians. The thick mucus from the adenoid, post nasal drip flowing down the posterior pharyngeal wall, may settle on the vocal folds and burden them and cause additional vocal effort. Allergy rhinitis may be important in this regard, but more than 60% of children in the study have not been tested for it. Therefore, some clinicians recommend the inclusion of antihistamine drugs in young patients treated for vocal nodules.
In addition, we would like to point out that in our previous studies we showed a correlation between adenoid mucus and the thermal season. To exclude this relationship, the control group was selected so that it did not differ statistically significantly from the study group in terms of the season.
- Since the majority of the children in the study group have allergies, it is essential to have information regarding chronic cough and the use of steroid inhalers, as these are also parameters implicated in the formation of vocal cord nodules.
Most of the children in both groups were not tested for allergies and therefore did not take antihistamines or steroids. However, if the allergy was studied, the group with vocal nodules had a more common allergy. All 6 children treated for asthma took inhaled steroids, but this group accounted for only 11% of children with nodules and only in this group should the effect of steroids on the formation of vocal nodules should be sought.
- Even if the authors provide an explanation implicating subglottic pressure in the discussion section, it is not rational to relate nodules to the absence of snoring. Snoring is a pathological sign. It would be more rational if the conclusion was that snoring is not related to vocal cord nodules, which would be interesting.
We’ve corrected it in the conclusion. Thank you.
- To study the hypothesis that “a small adenoid may reduce veloadenoidal closure, forcing an increase in subglottic pressure,” the authors should better compare two random groups of children with adenoid hypertrophy and without any vocal cord lesions. In this study, where one group of children already has nodules, the results are biased.
Certainly, adding a third control group – without nodules and symptoms of adenoid hypertrophy could be helpful in this regard, but it was not planned during the study. The initial hypothesis that we planned to test was to confirm that impaired nasal patency affects the formation of vocal nodules, so we compared the study group with the group with adenoid hypertrophy symptoms. The result was a surprise to us, as it is commonly believed that breathing dry air through the mouth promotes the formation of vocal nodules. I hope that the results obtained will contribute to further research in this area.
- In line 180, page 5/9, what is the optimal adenoid size? Would it be normal for a child of about 5 years old to have no adenoids at all?
This is academic question what is the optimal adenoid size? because in preschool children without symptoms of adenoid hypertrophy, we also observe a different size of enlarged adenoid tissue. When analyzing the available scales for the adenoid assessment, most of them classify the degree of hypertrophy, not the normal value and the value above which hypertrophy is defined.
Thank You
Reviewer 3 Report
Comments and Suggestions for Authors
well planned and well executed and presented protocol . Hoping you will continue exploring correlations with voicing attitude and possible impairment of audiophonatory reflex on useless voicing intensities . (BTW at line 132 it seems to me you meant "that" instead of "what" : am I wrong?)
Author Response
Der Reviewer,
Thank you very much for the huge amount of work put into evaluating my work and trying to make it better. Below I will try to give a precise answer to the questions posed
line 132 it seems to me you meant "that" instead of "what" : am I wrong?)
Corrected -Thank you
Round 2
Reviewer 2 Report
Comments and Suggestions for Authors
Dear authors thank you for comments in my review.
- Please highlight your comment in the text that you have performed a bilateral endoscopy.
- What was the reason for the children having thick mucus on the adenoids? Were they assessed during the course of an infection? Or was this mucus attributed to an allergy to LPR? All three of these parameters could account for the horseness of the children, and the mucus in that case could be a consequence of these conditions, rather than an independent factor related to the formation of the nodules. Since we don’t have information regarding these parameters, this jeopardizes the scientific significance of the relation of postnasal mucus with vocal nodules.
- I had previously discussed regarding the discussion and conclusion section that even if the authors provide an explanation implicating subglottic pressure in the discussion section, it is not rational to relate nodules to the absence of snoring. Snoring is a pathological sign. It would be more logical if the conclusion was that snoring is not related to vocal cord nodules, which would be interesting. The authors wrote that they corrected that, but it is still not changed in the second version of their manuscript that has been provided to us. Actually, the two versions are identical.
- To study the hypothesis that “a small adenoid may reduce veloadenoidal closure, forcing an increase in subglottic pressure,” the authors should better compare two random groups of children with and without adenoid hypertrophy and weighted for other parameters, especially the voice abuse level and examine whether they have any vocal cord lesions. In this study, where one group of children already has nodules, the results are biased.
- In line 180, page 5/9, the authors are, actually, referring to an optimal adenoid size, this is why I am asking what is the optimal adenoid size from their point of view.
- Overall, the two group samples are not weighted for the most important parameter for vocal cord nodules formation, the voice abuse. You have a group of children with voice disorders due to nodules and probably voice abuse and you compare it with a group that came to be examined due to symptoms of nasal obstruction typically by enlarged adenoids, who do not necessarily have voice abuse and the result is that small adenoids may be a factor for vocal nodes formation. The results do not seem to be scientifically sound.
Author Response
Dear Reviewer,
Thank You for your fast response and recommendations,
- Please highlight your comment in the text that you have performed a bilateral endoscopy.
Q: What was the reason for the children having thick mucus on the adenoids? Were they assessed during the course of an infection? Or was this mucus attributed to an allergy to LPR? All three of these parameters could account for the horseness of the children, and the mucus in that case could be a consequence of these conditions, rather than an independent factor related to the formation of the nodules. Since we don’t have information regarding these parameters, this jeopardizes the scientific significance of the relation of postnasal mucus with vocal nodules.
R: Mucus retention on the adenoid is one of the parameters we use for endoscopic assessment of the pharyngeal tonsil in children (Masna scale https://www.mdpi.com/1457826 ). Such retention can be related to allergies, but also LPR. However, you should be aware that allergy in children may manifest itself / be confirmed in tests at a later stage, when there is a sufficiently strong stimulation of the allergen, especially if these are outdoor allergens. Yes, also in our opinion, the flow of secretions and settling on the vocal folds promotes the formation of nodules. In our study, we showed that children with nodules have more mucus on the adenoid and therefore more post nasal drop to the pharynx.
Q: I had previously discussed regarding the discussion and conclusion section that even if the authors provide an explanation implicating subglottic pressure in the discussion section, it is not rational to relate nodules to the absence of snoring. Snoring is a pathological sign. It would be more logical if the conclusion was that snoring is not related to vocal cord nodules, which would be interesting. The authors wrote that they corrected that, but it is still not changed in the second version of their manuscript that has been provided to us. Actually, the two versions are identical.
R: I'm very sorry, there was an error in the attached version, I hope that this time we will be able to attach the correct one with corrected conclusions.
Q: To study the hypothesis that “a small adenoid may reduce veloadenoidal closure, forcing an increase in subglottic pressure,” the authors should better compare two random groups of children with and without adenoid hypertrophy and weighted for other parameters, especially the voice abuse level and examine whether they have any vocal cord lesions. In this study, where one group of children already has nodules, the results are biased.
R: The problem of voice abuse is undoubtedly very important in the formation of vocal nodules. Yes, during the visits, we analyze whether the child is expressive, shouts, has older/younger siblings, or goes to kindergarten. However, the assessment is very difficult to compare, in our group there are, for example, calm boys who abuse their voices only during sports activities, children who behave calmly at home and are very loud in kindergarten, or children with ADHD.
Q: In line 180, page 5/9, the authors are, actually, referring to an optimal adenoid size, this is why I am asking what is the optimal adenoid size from their point of view.
R: This is a very difficult question that you are asking me again, because it has been bothering me for many years and is an area of my research. From the point of view of adenoid symptoms – snoring, sleeping with the mouth open, frequent upper respiratory tract infections, tonsil size on the A/C ratio of 70% and more is a clinical problem, and adenoid with an A/C ratio of 80% should be treated surgically. If we assess the adenoid, for example, in terms of OME, the mucous on adenoid is more important than the size itself. However, in terms of optimal veloadenoidal closure in preschool children, it seems that the adenoid size should contribute about 70%, but this requires further research in this area, to which I hope this work will contribute.
Q: Overall, the two group samples are not weighted for the most important parameter for vocal cord nodules formation, the voice abuse. You have a group of children with voice disorders due to nodules and probably voice abuse and you compare it with a group that came to be examined due to symptoms of nasal obstruction typically by enlarged adenoids, who do not necessarily have voice abuse and the result is that small adenoids may be a factor for vocal nodes formation. The results do not seem to be scientifically sound.
R: I cannot agree with this statement because, as I pointed out earlier, it is very difficult to assess the expression of children, and in our observations, there are both children with ADHD and hypertrophic adenoid without vocal nodules, as well as calm boys who occasionally abuse their voice, in whom we find vocal nodules. Vocal trauma is certainly important, but there are other factors worth considering. However, don't you wonder about the fact that vocal nodules have formed in patients undergoing adenoidectomy, they are still the same screaming children?
In addition, this work indirectly verifies some previous theses, such as the one given by some authors that breathing dry, unhumidified air (breathing through the mouth) promotes the formation of vocal nodules. Therefore, we compared our group with children with adenoid hypertrophy in whom the oral respiratory tract predominates, it turned out that in children with nodules there is better nasal patency, they breathe humidified air and yet nodules form. This work is therefore intended to contribute to further analyses and research.
Sincerely Yours
Aleksander Zwierz